# Probing the rules of cell coordination in live tissues by interpretable machine learning based on graph neural networks

**Takaki Yamamoto**[1]*, **Katie Cockburn**[2,3], **Valentina Greco**[2,4], **Kyogo Kawaguchi**[1,5,6]*

**1** Nonequilibrium Physics of Living Matter RIKEN Hakubi Research Team, RIKEN Center for Biosystems Dynamics Research, Kobe, Japan, **2** Department of Genetics, Yale School of Medicine, New Haven, Connecticut, United States of America, **3** Department of Biochemistry and Rosalind & Morris Goodman Cancer Institute, McGill University, Montreal, Quebec, Canada, **4** Departments of Cell Biology and Dermatology, Yale Stem Cell Center, Yale Cancer Center, Yale School of Medicine, New Haven, Connecticut, United States of America, **5** RIKEN Cluster for Pioneering Research, Kobe, Japan, **6** Universal Biology Institute, The University of Tokyo, Tokyo, Japan

* yamamototakaki1212@gmail.com (TY); kyogo.kawaguchi@riken.jp (KK)

## Abstract

Robustness in developing and homeostatic tissues is supported by various types of spatio-temporal cell-to-cell interactions. Although live imaging and cell tracking are powerful in providing direct evidence of cell coordination rules, extracting and comparing these rules across many tissues with potentially different length and timescales of coordination requires a versatile framework of analysis. Here we demonstrate that graph neural network (GNN) models are suited for this purpose, by showing how they can be applied to predict cell fate in tissues and utilized to infer the cell interactions governing the multicellular dynamics. Analyzing the live mammalian epidermis data, where spatiotemporal graphs constructed from cell tracks and cell contacts are given as inputs, GNN discovers distinct neighbor cell fate coordination rules that depend on the region of the body. This approach demonstrates how the GNN framework is powerful in inferring general cell interaction rules from live data without prior knowledge of the signaling involved.

**Data Availability Statement:** All data needed to evaluate the conclusions in the paper are present in the paper and/or the Supplementary Materials. The pipeline of the GNN framework as well as the

## Author summary

During development and homeostasis, cells coordinate with each other to grow, deform, and maintain the tissues. Even with the modern high-throughput cell profiling technologies and high-resolution microscopy, it is still challenging to infer how cell coordination affects the dynamics such as cell fate choice, due to the complexity of the problem and the limited methods to perform perturbation experiments. We here propose a versatile framework of analysis utilizing an interpretable machine learning method based on graph neural network (GNN) which infers the cell-to-cell interaction rules from live images of multicellular dynamics. From the spatiotemporal graphs generated from live images of skin stem cells, we identified previously unaddressed neighbor fate coupling as well as rules consistent with past findings. We further found distinct interaction rules in a

analyzed data (tracks of cells in the epidermis and numerical simulations) is available on a GitHub repository at https://github.com/ NoneqPhysLivingMatterLab/cell_interaction_gnn.

**Funding:** This work is supported by the funding from JSPS KAKENHI grants number 19K16096 (T. Y), JP18H04760, JP18K13515, JP19H05275, JP19H05795, and Research Grant from Human Frontier Science Program (Ref. Grant No. RGY0081/2019) (K.K), HHMI Scholar award (55108527) and NIH grants number R01AR063663, R01AR072668, DP1AG066590 (V. G), the Canadian Institutes of Health Research, and as a New York Stem Cell Foundation Druckenmiller Fellow (K.C.). The funders had no role in study design, data collection and analysis, decision to publish, or preparation of the manuscript.

**Competing interests:** The authors have declared that no competing interests exist.

different skin region of the body, indicating that our method is useful in probing the diverse mechanism behind the robustness and flexibility in multicellular systems. The GNN framework is applicable for interaction rule discovery for general multicellular dynamics as well as in a wide range of systems where modeling by stochastic interacting agents is effective.

## Introduction

Robustness in developing and homeostatic tissues is supported by the spatiotemporal cell-to-cell interactions, i.e., feedback mechanisms acting at various time and length scales that prevent overgrowth or depletion. In the stem cell pool of homeostatic tissues, the spatial aspect of feedback have been shown to range from direct cell contacts [1–4] to more indirect and longer range forms such as niche competition [5, 6]. The fate correlation in time can come in different forms; growth and loss of cells may occur almost simultaneously within the tissue, or there could be regeneration cycles with longer time scale as in the case of hair follicles [7]. Moreover, the ordering of the events in the feedback scheme can be different; cell divisions can compensate for earlier cell loss, or vice versa.

Although there are promising attempts to probe the cell-to-cell interactions from high-throughput single-cell analyses [8–14], live imaging of tissues followed by accurate tracking of the cells is still the most direct method to probe the details of the spatiotemporal feedback [15]. In the case of the mouse hind paw epidermis, analysis of the live images and cell tracks led to the finding that the skin stem cells have coordinated fate between the nearest-neighbors within a short time frame; cell divisions were coordinating with neighboring cell delamination (due to differentiation) with 1–2 days delay [1]. However, whether such short spatiotemporal range of coordination as well as the time-ordering found in the hind paw epidermis is adopted in other tissues is currently unknown. In fact, the detailed rules of coordination and feedback have not been tested even for mouse skin regions other than the hind paw.

To compare the cell coordination rules adopted in different tissues, it will be desirable to have a generic framework of analysis where such rules can be suggested unbiasedly, rather than preparing methods specific to each tissue. To this end, we may train a model based on machine learning to predict the future behavior of cells from past data, and later challenge the model to extract the rules that have been learned by the machine. The input to this machine can include high-content intracellular features such as the cell morphology and gene expression, as well as the interaction between cells. The key in this procedure will be to design a versatile machine learning scheme that gives predictions of the future dynamics while retaining high interpretability: the ability to partially mask data, apply attribution methods [16], or use symbolic reductions. Such method will be in contrast to hypothesis-based approaches where certain coordination rules cannot be inferred by design. For example, the analysis conducted to extract the coordination features in the hind paw epidermis [1] assumed cell behavior inducing other cell's activity (i.e., cell division or delamination), which cannot extract suppressive effects such as induced silencing of neighboring cells [17, 18].

Machine learning methods have been heavily used to extract features of cells from live or fixed cell image sequences. Applications include segmentation and tracking [19–21], in silico labeling [22], and direct prediction of cell fates [23], which typically use versions of convolutional neural networks (CNN). The downside in using image data directly for cell fate prediction is that images contain redundant information making it difficult to focus on the most relevant components to interpret the rules. For instance, a significant feature of cells

undergoing cell division is their size growth, which means that the machine will likely learn through images that cell size is a good predictor of cell fate. This strong association will be a problem when we are interested in the mechanism upstream of fate decision (i.e., commitment to cell division), since cell size is difficult to mask out from images without affecting the other components. On the other hand, important aspects such as cell tracks, lineages, and contacts are not trivially deducible by the machines from the images, making it harder to conduct interpretable rule extractions based on the experiences of individual cells. Other methods that have been proposed to infer the rules of multi-component dynamics by interpretable machine learning frameworks [24] currently do not have the resolution that enables the rule extraction at the level of individual cells.

A promising machine learning approach toward automatic cell interaction rule discoveries is the graph neural network (GNN) scheme [25]. In GNN models, the structure of a graph can be taken as an input as well as the features associated with each node, which gives flexibility compared with conventional methods such as CNNs. Taking advantage of its interpretability, the GNN approach has succeeded in forward dynamics predictions of physical systems [26] as well as in inferring the rules of agent-interactions [27] and dynamical properties [28] from timelapse data obtained by simulations and experiments. The GNN framework is particularly suited for representing the heterogeneous interactions between cells as well as the intracellular features extractable from image data such as cell shapes and gene expressions [29]. The dynamical features of multicellular systems can be captured by further incorporating the information of cell tracks and constructing a spatiotemporal graph. However, while GNN models have been proposed to deal with time-evolving graphs with nodes and edges being added or removed [30–32], there is still no example where replicating nodes are taken into account, which is important in describing cell divisions.

In this work, we propose a GNN-based framework that incorporates the cell interactions as well as the cell tracks including divisions, and demonstrate how the framework can be applied to novel data to extract and compare the different cell fate coordination rules across tissues. First, we observe that a GNN model constructed from cell contacts and cell lineage can predict the fates of the mouse skin stem cells, indicating that the graph structure encodes significant information of cellular dynamics. Next, we demonstrate how the rules learned by the GNN model can be extracted by ablating the input features, re-configuring the model step-by-step, and employing the attribution method. As a result, we find neighboring cell fate coordination rules that have been overlooked in the paw skin data, as well as rules that are consistent with the previously conducted fate imbalance analyses [1]. For our main application, we analyze the novel tracking data of the ear skin stem cells [33], and find an additional coordination rule that is absent in the paw skin. We further confirm the validity of the method by applying it to data generated by numerical simulations that encode stochastic cell dynamics mimicking the homeostatic epidermis. Together, this work demonstrates how the GNN model can be applied to highly stochastic kinetics with agent loss and gain, and how it can be utilized in solving the mechanism of multicellular kinetics systematically.

## Results

### Data and construction of GNN

The epidermis is maintained by continuous cell divisions occurring in a pseudo-two-dimensional region called the basal layer. Cells in the basal layer can irreversibly delaminate toward the suprabasal layer and differentiate (i.e., turn post-mitotic), and eventually shed off (Fig 1A). We use the dataset of cell tracks previously generated from live images collected from the non-hairy mouse plantar (hind paw) skin [1], which includes the tracks of all the cells within a

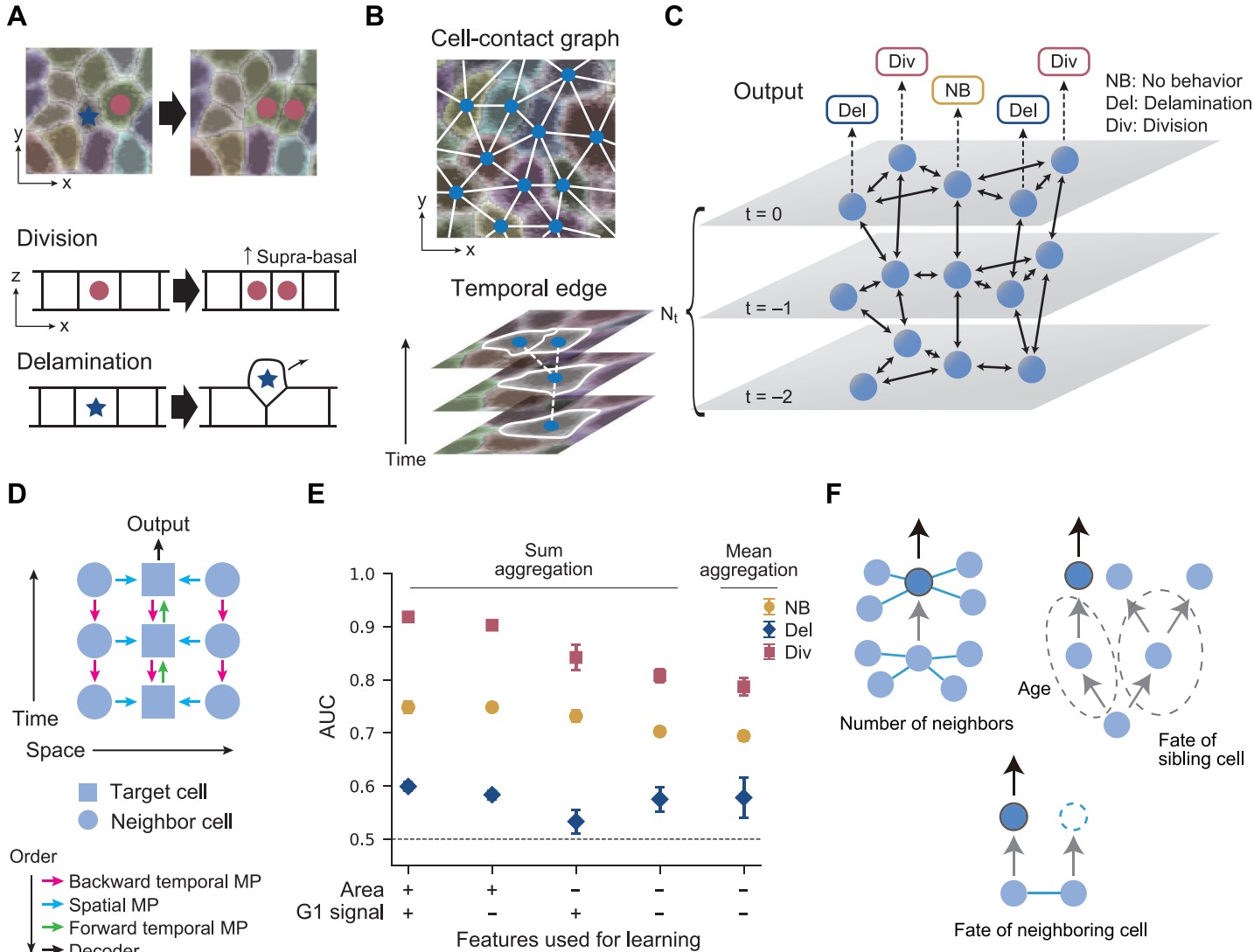

**Fig 1. Bidirectional spatiotemporal GNN model: Analysis of the hind paw epidermis data.** (A) Schematic of cell division and delamination in the basal layer of the mouse epidermis. (B) Schematic of the cell-contact graph and temporal edges in the basal layer. (C) Schematic of the 2+1 dimensional spatiotemporal graph. A 3-time model is shown as an example. The spheres represent cells, and double-headed arrows represent the neighboring cell connection and the temporal tracks. The GNN models predict the cell fate ($t = 0$). (D) Schematic of the information flow in the bidirectional spatiotemporal GNN model. The information flow in a target cell's subgraph is shown. The target and the neighboring cells are represented by boxes and circles, respectively. Each arrow indicates the direction of the information flow, the order of which is represented by the different colors. (E) The area under the curve score (AUC) of the bidirectional spatiotemporal GNN models with sum and mean aggregation is shown for models under various feature conditions. The AUC for each cell fate label is shown with different markers (NB: circle, Del: diamond, Div: square), which are obtained by averaging over six trained models. Error bar: standard deviation. (F) Graph motifs that are potentially useful in predicting the target cell fate. The correlation of these motifs to the target cell fate can be exploited to make predictions in the bidirectional spatiotemporal GNN model.

region of the basal layer. Between the time frames, cells conduct one of the three possible behaviors; divide (Div), delaminate from the basal layer and migrate into the suprabasal layer (Del), or stay (no behavior, NB), typically with the ratio of 1:1:8 within the time interval of the original image acquisition (12 hours). Loss by cell death is negligibly rare in this tissue under homeostasis. The relative motion between the cells is slow, meaning that the main source of the cell motion is displacement due to cell division and cell delamination.

In our GNN framework, we represent the spatio-temporal dynamics of the tissue by graphs designed to capture 1) the spatial relationship including the cell contacts as well as the lineages and events such as cell delamination and division, and 2) any cell feature such as the gene expression levels and cell area in the basal layer. We first construct graphs with the basal layer cells as nodes, connected by edges in spatial and temporal directions. The spatial edges represent the cell-to-cell contact obtained from the two-dimensional segmentation in the basal layer, based on the membrane reporter signal (Fig 1B and S1 Video). The temporal edges are the tracks of the cells, involving forks corresponding to cell divisions and terminal ends representing cell delamination; we do not track the cells after they have left the basal layer. In each node (i.e., cell in a time frame), multiple features extracted from the raw images can be assigned. For the hind paw data, we chose the cross-section area of the cell in the two-dimensional basal layer calculated from the segmentation, and the level of the Fucci-G1 reporter [34] which is the integral of the corresponding fluorescent channel within the segmented cell area. The GNN model takes in the spatiotemporal graphs as well as the features associated with each node, and outputs predictions on the behavior of cells in the last frame (Div, Del, or NB).

We trained a GNN model using training and test data taken from two separate regions of the same mouse hind paw, each involving more than 200 cells tracked over fifteen time frames with a total of approximately 250 Div and Del events occurring in each region. In the training of a $N_t$-time model, we extracted graphs composed of sequential $N_t$ time frames, where the time frames are indexed from $t = -(N_t - 1)$ to 0 in temporal order (Fig 1C), and conducted supervised machine learning to predict the fate of the target cell in the last frame $t = 0$.

The model calculates the output from the input by sequentially updating the features in the nodes in the following way (information flow schematically shown in Fig 1D: bidirectional spatiotemporal GNN model). First, the model performs message passing (MP) on the temporal edges from the future to the past $N_t - 1$ times. Here, a single MP is conducted by concatenating the features of adjacent nodes to be processed through a multi-layer perceptron (MLP), and then passing on that processed feature to the past node. Second, the model performs MP bidirectionally on the spatial edges $N_s$ times (Fig Ai in S1 Appendix). Third, the model performs MP $N_t - 1$ times on the temporal edges from the past towards the future nodes. The order of the MP (Fig 1D) and the sum aggregation method [35] are set this way so that the spatiotemporal graph structure is correctly reflected in the output. In the end of the MP, the node feature in the last frame incorporates the information from its $N_t$-frame ancestor nodes, $N_s$-step neighbor nodes of the ancestor nodes in the cell contact graph, as well as the daughter cells of the ancestor neighbors (Fig 1C). We fix $N_s = 1$ in the following analyses since the performance of the model were similar for $N_s = 1$ and 2 (Fig A in S1 Appendix), and also $N_t = 4$, which will be changed later. Finally, we decode the node feature in the last frame via another MLP and output the softmax score for each cell fate label (Div, Del, and NB) to conduct three-class classification. Here, we chose MP rather than convolutional [36, 37] or attention-based GNN [38] because MP has higher flexibility in representation. The generic functional form described in MP may become important to probe complex nonlinear interactions, for instance when high-dimensional data such as gene expression and morphology are used as cell features.

## Spatiotemporal cell interaction graph predicts fate without cell features

The GNN model was successfully trained using standard methods employing a loss function (softmax-cross-entropy loss for the three cell behavior labels) and the Adam optimizer (Fig B in S1 Appendix). The Area Under the Curve (AUC) score shows that all behaviors are predicted significantly better than a random guess (AUC = 0.5) (Fig 1E). Also, the small standard deviations between the training samples indicate the trained models are reproducible. In

particular, we found that cell divisions can be predicted with high accuracy compared with Del and NB. We further found that by training the model with reduced features (i.e., without the cell area, G1-phase reporter signal, or both), the score decreased significantly, indicating that these features are utilized in making reliable predictions. Indeed, the cross-section area tends to be larger for dividing cells (Fig Ci and v in S1 Appendix), consistent with previous observations [1, 39]. Furthermore, the G1-phase signal tends to be lower in dividing cells as expected, while there was no difference in the G1-phase signal between NB and Del cells (Fig Cvi in S1 Appendix).

Intriguingly, we found that even without the node features, the GNN model can predict the fates with significant scores (Fig 1E). This indicates that the graph structure itself encodes useful information in predicting cell fate. A candidate structure in the graph that can be used in the prediction is the number of edges (i.e., neighboring cells, Fig 1F). Indeed, the number of neighboring cells is positively correlated with cell division in the next time frame, which is explained by the fact that cells with a larger area tend to be in contact with more cells in the basal layer (Fig Ciii in S1 Appendix). To see whether the number of edges is important in the prediction, we changed the model to take mean aggregation in the calculation step of the spatial edges, which will make the number of contacts invisible in the model. We found that this change in the model only has a minor effect on the AUC score, suggesting that there are other subgraph motifs that are utilized in the prediction.

Other subgraph motifs which can be interpreted in the biological context include the temporal length up to the last cell division point (Fig 1F). This is essentially the age of the target cell, which can be used to predict fate if the fate choice is temporally non-random [40]. The neighbor cell fates are also reflected as motifs, which should be important according to previous results [1]. Similarly, the sibling cell behavior can be exploited if there are sibling fate correlations such as in the case of asymmetric division [41], although this is less relevant for the current dataset [1, 40].

## Reduced GNN model and attribution scores reveal cell interaction rules in the hind paw and ear epidermis

In order to identify which subgraph motifs are responsible for the predicted future cell dynamics, we next considered reducing the GNN model to observe how the predictability decreases. By simply excluding the initial step of propagating the information toward the past frames, the information from the temporal branches in the sibling and neighboring cells (including their fates, Fig 1F) will not be transmitted to the target cell (Fig 2A: unidirectional spatiotemporal GNN model). To add-back the information of neighboring cell fate, we introduced an additional feature to each cell, the next frame behavior (NFB), which takes Div, Del, or NB (Fig 2B). Importantly, this additional feature was added only to the past cells ($t \leq -1$), since the NFB of the cells at $t = 0$ is the target of prediction.

The new GNN model predicts cell fates almost as well as the previous model with and without the area and G1-phase reporter signal features (Fig 2C and Fig D in S1 Appendix), indicating that the key ingredients are captured without the backward-time propagation as far as the neighboring cell fates (NFB) are added back. By further eliminating the NFB and seeing that the predictability drops, we confirmed that this feature indeed has impact on the prediction of all behaviors (Fig 2C and Fig E in S1 Appendix).

To further address the detail of the mechanism of prediction, we next employed the attribution method called the integrated gradient (IG) [16]. In this method, the impact of each input variable on the output is quantified by integrating the change in the output score level upon the gradual shift of each input feature (Fig 2D and 2E and Fig F in S1 Appendix). The IG also

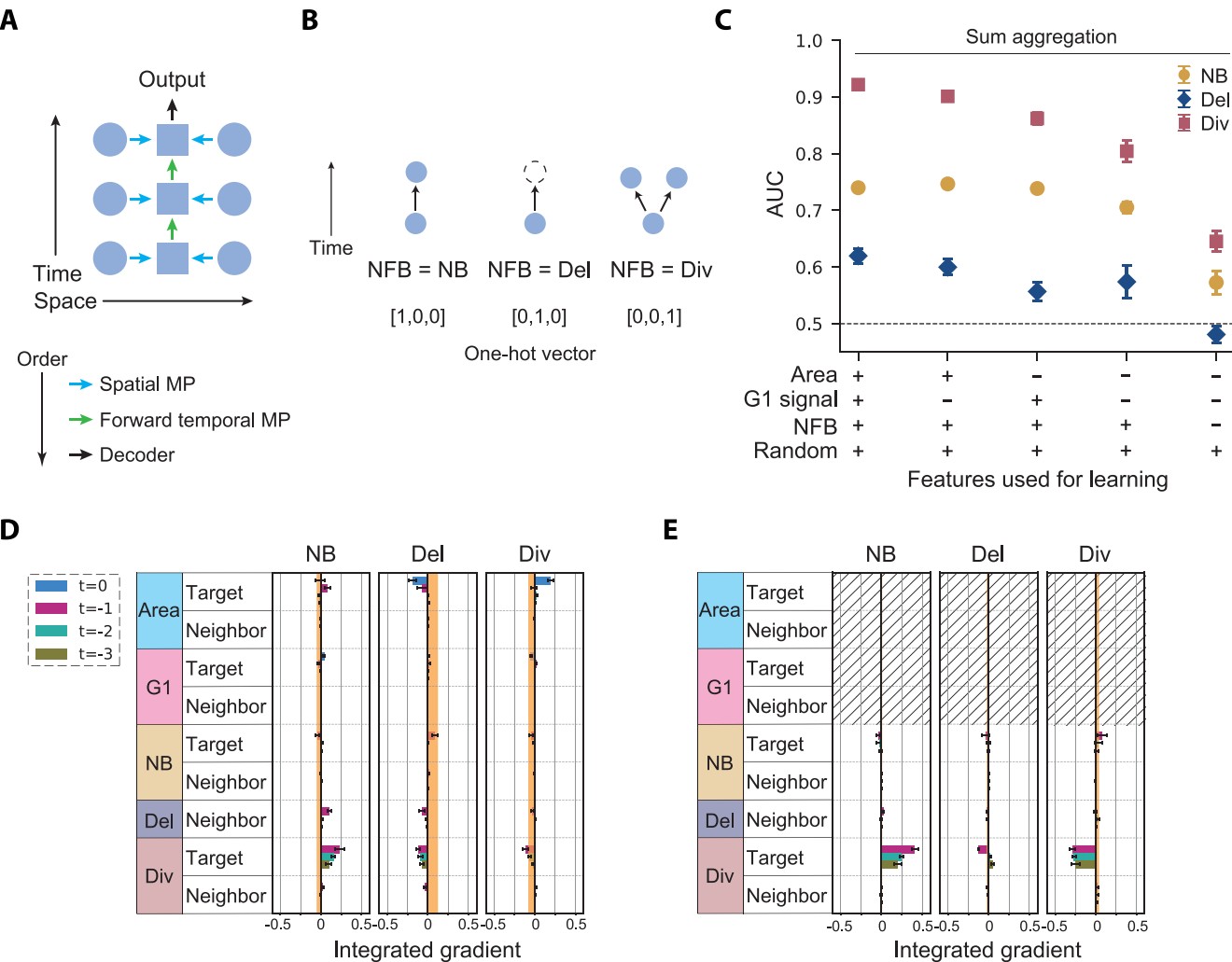

**Fig 2. Unidirectional spatiotemporal GNN model with sum aggregation.** (A) The next frame behavior (NFB) feature introduced to represent the local branch structure. (B) Schematic of the information flow in the model of the unidirectional spatiotemporal GNN (see Fig 1D). (C) The AUC of the unidirectional spatiotemporal GNN is shown for models under various feature sets, obtained by averaging over six trained models. Error bar: standard deviation. (D and E) The attributions of the unidirectional spatiotemporal GNN model are shown for each feature condition: (Area, G1-phase signal, NFB, Random) = (+, +, +, +) (D) and (−, −, +, +) (E). The integrated gradients (IG) averaged over six trained models is shown for each pooled feature. Positive and negative IGs represent the positive and negative effects of the features (rows) on the fates (columns), respectively. Error bar: standard error. The upper and lower values of the IGs of the random features are shown as the orange zone as an indicator of the baseline IG level.

reflects the sign of the impact of each feature; a negative attribution score of feature *f* toward fate *a* means that the observation of *f* will decrease the output score towards predicting *a*. The AUC and the IG scores are complementary to each other; the IG highlights the significance of inputs at a finer scale than the AUC, but does not give quantitative values of predictability. To account for the noise in IG induced by finite sample size, we further added a random feature to the cells, i.e., uniform random numbers ranging from 0 to 1 that have no correlation with cell fate (Fig Dii in S1 Appendix), and calculated its IG score to identify the baseline for non-zero signal (orange shades, Fig 2D and 2E and Fig F in S1 Appendix).

First, in Fig 2D and Fig Fi in S1 Appendix, the area of target cells at *t* = 0 are found to give positive and negative attribution for the prediction of cell division and delamination, respectively. This indicates that cells with large areas tend to conduct cell division in the next frame,

while they are unlikely to delaminate. GNN models trained without spatial MPs also show that the area of target cells has indeed significant predictability of the cell fates (Fig G in S1 Appendix). On the other hand, in Fig Fii in S1 Appendix, negative attribution is observed in the G1-phase signal of the target cells at $t = 0$, which is consistent with the expectation that a high G1-phase signal of the target cell should predict no cell division in the next frame. Interestingly, the G1-phase signal has a relatively low score when the cell area feature is present (Fig 2D), indicating that the G1-phase signal is mostly redundant for cell fate prediction. These results highlight how the model is efficiently focusing on the important variables from the multi-dimensional input.

Without the cell area and G1-phase reporter signal, non-zero attribution scores are found for the prediction of fates by the division of the target cell (Fig 2E). This is reflecting how the age of the cell can be used in the cell fate prediction; newly born cells tend to undergo a refractory period [40], meaning that the target cell is less likely to divide again if it divided (was born) recently. More generally, the distributions of lifetime can be distinct between cells with different fates (Fig H in S1 Appendix), which can be exploited in making fate predictions. To eliminate the cell age information altogether, we set all the features of the target cell sequence to zero (Fig 3A: cell external model). We also took mean aggregation in conducting the MP on spatial edges to make the number of contacts invisible. This new model can still learn to predict the fates significantly better than random (Fig 3B). We found a high IG score indicating that Del of a neighboring cell 24 hours before (two frames prior) is useful information in predicting Div. Consistent with this, when changing the number of input time frames $N_t$, the AUC score largely increased between $N_t = 2$ and 3 (Fig Ii in S1 Appendix). These results match with the previous observation that cell delamination correlates with neighboring cell division 1–2 days later [1].

Moreover, the GNN model predicts that there is an effect of neighboring Del suppressing Del in the next frame. This indicates that cell delamination is not entirely random, and there may exist a mechanism to suppress two or more neighboring cells to delaminate at the same time. This finding demonstrates that the rule extraction procedure using the GNN model is useful in predicting unexplored mechanisms. Importantly, while the AUC is low for some cases due to the high stochasticity in the cell fate events, it is still possible to determine the fate correlations in neighboring cells.

We next analyzed the recently obtained mouse ear skin data [33] to compare the mechanism of homeostasis across different tissues. Although the ear skin has more structure (i.e., hair follicles and other appendages), we have previously observed that the behaviors of the interfollicular epidermal basal cells in the ear skin are similar to that of the hairless hind paw [1, 40]. A noticeable difference is that the rate of divisions and delaminations were approximately two-fold slower in the ear, which is why the interval between the frames was set as 24 hours for the ear data acquisition. For our purpose, we generated a novel dataset of whole region cell tracks from 6 regions in the ear epidermis with 54 time frames in total.

In Fig 3B and 3C, we show the AUC and the attributions for the ear data; the largest positive (negative) IG score for Div (Del) is attributed to the Del of neighboring cells from one frame earlier, indicating the existence of the same rule as the hind paw epidermis. Negative correlation between neighboring cell fates was expected to exist in the ear as well as it is critical in maintaining the stem cell pool [1]. However, the actual coupling as well as the ordering of the events have not been previously addressed.

In addition, there are large positive (negative) IG scores for Del (Div) attributed to the Div of neighboring cells, which were not seen in the hind paw epidermis. These scores imply that a mechanism of neighboring Div-induced Del and Div-suppression may exist. This difference was further confirmed by conducting a neighbor-fate imbalance analysis (Fig Ji and ii in S1

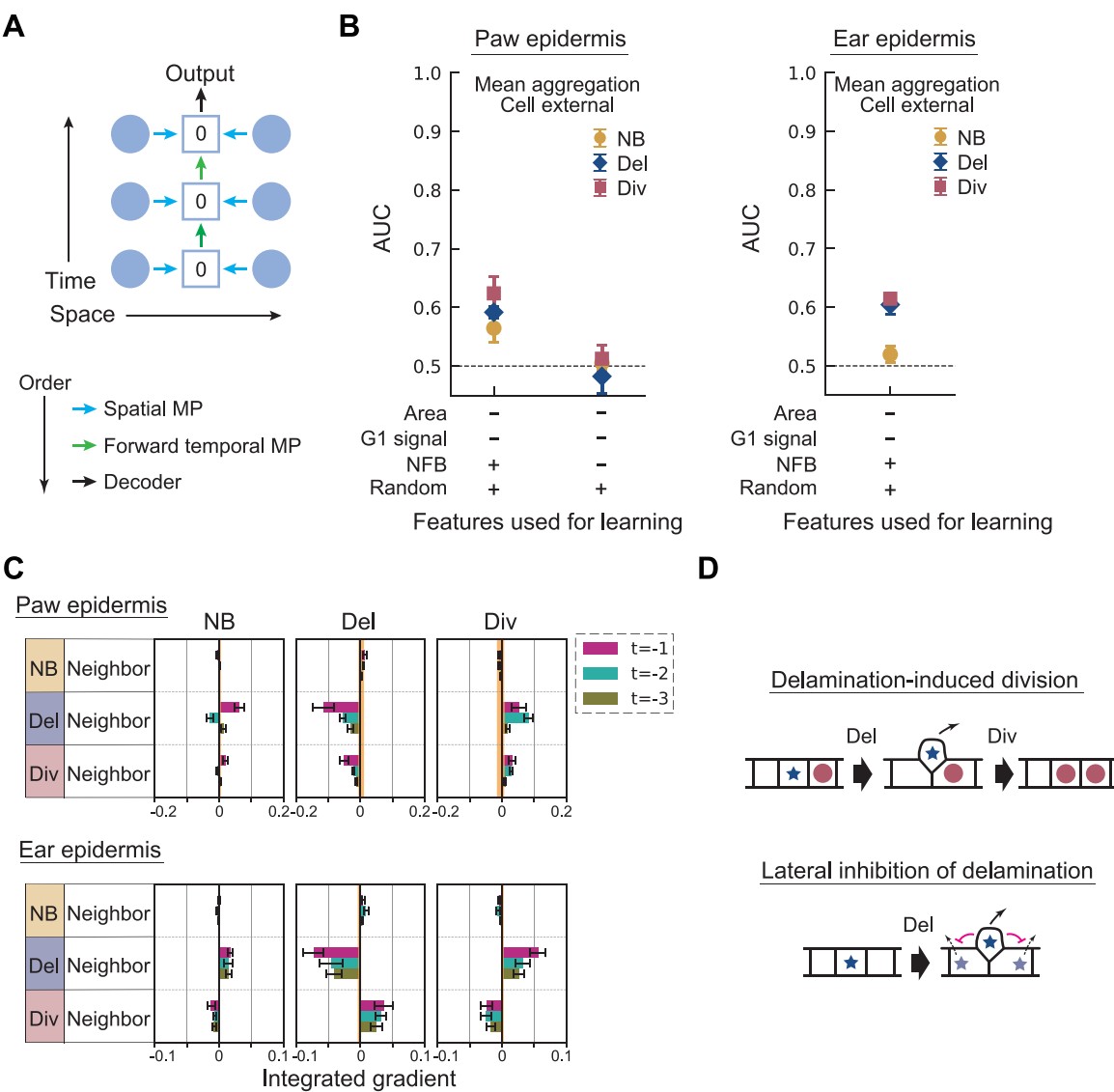

**Fig 3. Cell external model with mean aggregation applied to hind paw and ear epidermis data.** (A) Schematic of the information flow in the cell external model (see Fig 1D). The zero value in the target cell's box represents the null feature vector, which is assigned to the target cell's feature in the cell external model. (B) The AUC of the cell external model with mean aggregation is shown for models with and without future fate, obtained by averaging over six trained models (left: paw, right: ear). Error bar: standard deviation. (C) The attribution of the cell external model with mean aggregation is shown for the feature condition: (Area, G1 signal, NFB, Random) = (−, −, +, +) (top: paw, bottom: ear). The IG averaged over six trained models is shown for each pooled feature. Error bar: standard error. The upper and lower values of the IG of the random feature are shown as the orange zone. (D) Schematic of a delamination-induced division and lateral inhibition of delamination by neighbor cell delamination.

Appendix). In this analysis, we focus on individual cells that either divided or delaminated and follow the subsequent behaviors of their six-nearest neighbor cells to calculate the neighbor fate imbalance [1]. The average fate imbalance indeed deviates from zero for the neighbors of dividing cells in the ear, which is a distinct feature from the paw.

These results indicate that the GNN-based method can detect cell fate coordination as efficiently as the previous neighboring imbalance method [1] which was constructed specifically for this purpose. The neighboring imbalance method suffers from the subtlety of correcting the background temporal fluctuation of cell fates (i.e., some time frames have significantly

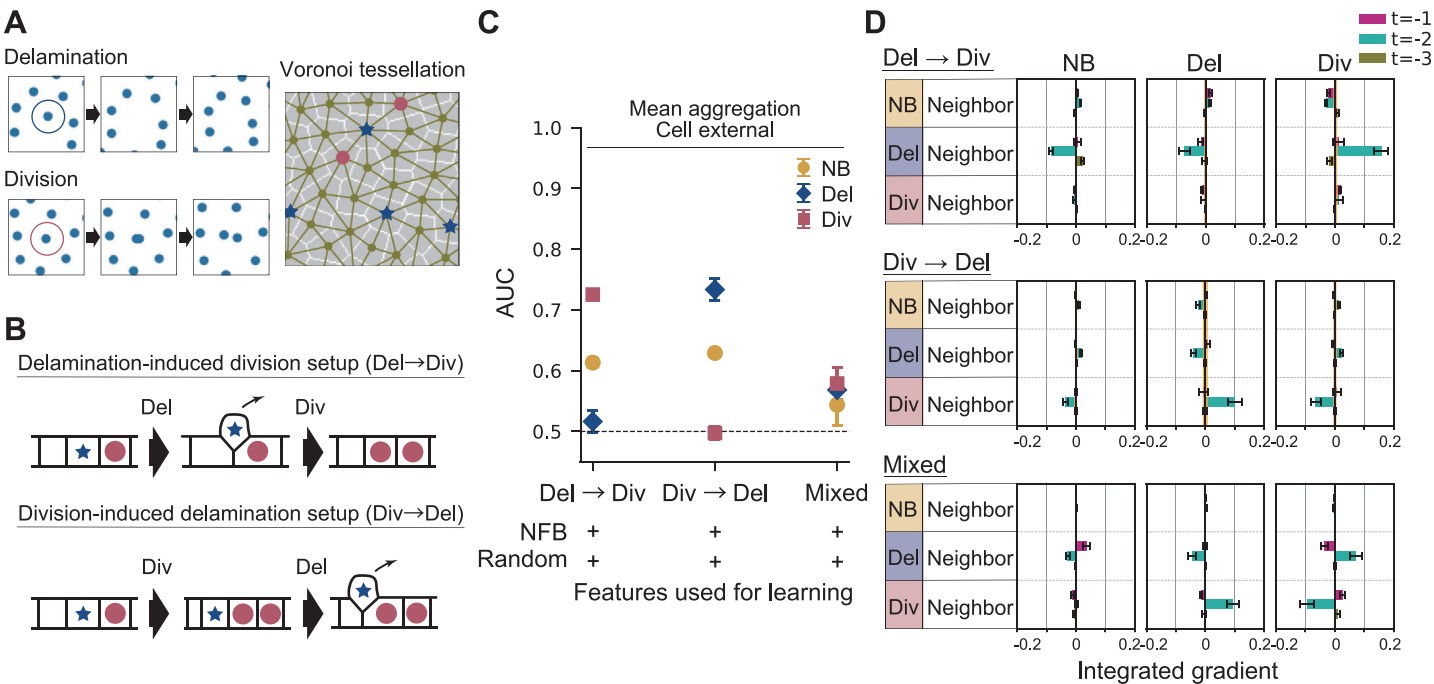

**Fig 4. Validation of the GNN approach by *in silico* models of homeostasis.** (A) Snapshots of simulation from the delamination-induced division setup. Cell contacts are defined by the Voronoi tessellation. Delaminating cells and dividing cells are represented by blue stars and red circle markers, respectively. (B) Schematic of the rules applied in *in silico* models of homeostasis: delamination-induced division, division-induced delamination, and mixed. (C) The AUC of the cell external model with mean aggregation is shown for the three *in silico* setups, obtained by averaging over six trained models. Error bar: standard deviation. (D) The attribution of the cell external model with mean aggregation is shown for three *in silico* setups. The IG averaged over six trained models is shown for each pooled feature. Error bar: standard error. The upper and lower values of the IG of the random feature are shown as the orange zone.

more Div of Del events than others), whereas for the GNN method the baseline is statistically controlled. For this reason, the method was sensitive enough to capture previously overlooked interactions of cell delamination suppressing neighboring delamination; this is not capturable by the neighboring imbalance analysis since it will only return a zero signal.

Here we demonstrated how the multi-step ablation strategy is useful in identifying the possible mechanisms behind cell fate determination. Starting from the full model, we first identified strong predictors of cell fate, such as the cell area, which were eliminated in the next model to seek further underlying factors. Although the causal relationships between the input variables are complex and difficult to disentangle in most cases, systematically eliminating the variables with high attribution score is a solid method to eventually reach the original cause.

### Cell interaction rule inference from numerical simulation data

Finally, we conducted numerical simulations of different coordinated cell fate models and tested whether artificially implemented rules can be correctly extracted. In the simulated models, cells are represented by points in the 2D space undergoing repulsive interaction with each other. Cell divisions and delaminations are recapitulated by abrupt point duplication (with small noise) and elimination, respectively (Fig 4A). We encoded the cell fate coordination in the following ways (Fig 4B). In the delamination-induced division setup, we first randomly picked a cell to delaminate and chose another cell randomly from its six-nearest neighbors to divide 24 hours after the first cell delaminated. In the division-induced delamination setup, the rules were flipped; a randomly picked cell was set to divide while one of its six nearest

neighbors was randomly chosen to delaminate after the division. We also tried another setup where these two rules were equally mixed.

Running the same GNN learning algorithm, we found that the AUC scores of each fate are in the expected order (Fig 4C); in the delamination-induced division setup, the division was predictable and delamination was not, and vice versa. By testing the attribution method on these simulated data, we confirmed that the inferred mechanisms indicate the implemented rules (Fig 4D). We further noticed that the predictability of the fates become significantly low for the mixed rule setup (Fig 4C), which is close to the case of experimental data (Fig 3B, C). Hence, the low score obtained in the experimental data prediction can be explained by the inherent stochasticity in the cell dynamics. Even in such situation, the attribution method was able to pull out the coordination rules. This indicates that the AUC does not necessarily have to be high in order to identify the correct interaction rules.

## Discussion

Here we have used the graph-based learning framework to systematically infer how interactions correlate with cell fates in the basal layer stem cell pool of the epidermis. The GNN models were able to predict the future cell fate using the relationships of the target cell and the cells in contact in the past time frames. We identified the cell features that are attributed to the predictability of the model using IG, and clarified that neighboring cell fates have a delayed effect on the target cell's fate outcome. The GNN-based approach gives interpretable attributions in the words of the spatiotemporal relationship between the cells, which is an advantage compared with direct image-based approaches employing CNNs.

The extracted rules from the dynamics in the epidermis included the previously reported delamination-induced neighboring division [1] as well as potentially novel interactions: suppression (induction) of cell delamination by neighboring delamination (division). Inhibition of certain behaviors by cells in contact is a common motif observed in cell biology [17, 42, 43]. Neighboring division-induced delamination may be associated with local mechanical forces [44]. Testing whether known molecular pathways responsible for lateral inhibition and mechanical signaling are playing roles in tissue homeostasis will be an interesting next step.

The predicted rules are favorable mechanisms in keeping the cell density in the homeostatic tissue, and it is intriguing that there exist differences in the rules adopted in distinct regions of the skin. A possible explanation for this is the cell density; the hind paw epidermis is crowded compared with the ear (26,000 cells/mm$^2$ versus 14,000 cells/mm$^2$, with almost the same thickness, 15 $\mu$m, Fig K in S1 Appendix). In high cell density regions such as the hind paw, it is reasonable that spontaneous cell division is suppressed and cells can only proliferate when there is space provided by neighboring delamination. Interestingly, in the developing skin where the cell density is much lower than the adult ear, the stratification of cells has been reported to be driven by neighboring cell division [45]. Given these facts, we hypothesize that the balance between cell differentiation and proliferation can be maintained through distinct mechanisms depending on the operating cell density regimes.

In predicting the kinetics, multiple time frames were required as inputs in our example (Fig I in S1 Appendix) due to the significant time delay between the cell fate events. This indicates that the proper treatment of the temporal axis was necessary, which is why GNN approaches based on static graphs [46–48] are not directly applicable to the problem of multicellular kinetics. Furthermore, our GNN framework takes into account the replication of nodes (i.e., cell division), which has not been addressed in previous models involving time-evolving graphs [30–32]. Since the incorporation of the cell tracks including cell division is fundamental in the

analysis of multi-cellular systems, this study serves as an important step towards building an unbiased rule extraction framework for tissue dynamics.

Multicellular dynamics is inherently stochastic owing to the complex interaction between cells and the environment as well as the single-cell level fluctuations. Likely due to this stochasticity, the prediction scores generated by the GNN in this study were far from accurate. Nevertheless, the models were able to provide sensible predictions and cell interaction rules from the epidermis data and inferred the correct rules from the simulation data. The success of the current approach on data generated by highly stochastic kinetics suggests that the framework is applicable not only for general multi-cellular dynamics such as developing embryos, malignant tissues, and organoids, but also for a wide range of systems where modeling by stochastic interacting agents is effective such as in disease spreading and ecology.

The current application relied on a relatively large data set of curated cell tracks from whole regions of the mouse skin. Although obtaining such data is still technically demanding in other biological tissues, recent microscopy and cell tracking methods are producing promising results in generating whole tissue level tracks in many systems [15]. In analyzing data for example from developing tissues, an important next step will be to develop a method to interpret the effects of higher-order subgraph motifs. We extracted the effects of neighboring cell fates by converting the cell fate motif (branches and termination) into a node feature variable, which will need to be generalized in order to capture more complex relations such as fate imbalance across generations and three-body cell interactions. In addition to the attribution method employed in this work, attention-based GNN [38] and other methods which attribute subgraph motifs to the prediction [49, 50] will be candidate strategies to improve the interpretability of GNN models and reveal complicated cell-to-cell interactions.

## Methods

### Data preparation

The cell track data of the mouse hind paw epidermis basal layer was generated in our previous work [1].

For the ear epidermis data, we used the images collected in the work of [33], and conducted the semi-automated tracking procedure similar to the previous method [1]. We first performed 3D segmentation by cellpose [21] using the nucleus channel (K14H2BmCherry) from a region size of 0.3 mm × 0.3 mm × 40$\mu$m and obtained the 3D masks of the cell nuclei. We then defined the height of the interface between the epidermis and the dermis based on the 3D masks of the nuclei and subtracted this height from the original 3D data to level the basal layer position. From the height-corrected 3D images, we took three consecutive z-positions containing the nucleus of all the basal layer cells and averaged the intensity over the three slices to obtain 2D images in each channel. We calculated the local maxima of the cell nuclei (K14H2BmCherry) and automatically corrected the shifts between time frames by minimizing the square distance between the nearest cell positions across the frames using affine transformation. The cells included in all ten time frames were used in the following analysis. The crop size in the 2D plane therefore varied across different areas, ranging from 139 $\mu$m × 139 $\mu$m to 238 $\mu$m × 238 $\mu$m.

At each time frame, we segmented the cells using the marker-controlled watershed algorithm using the mem-tdTomato channel and the maxima of the cell nuclei positions. We assigned each cell (i.e., segmented area) to a segmented area in the previous time frame with the largest overlap. Tracked cells were frequently lost or were associated with more than one cell in the subsequent time frame, which indicated cell delamination from the basal layer and cell division, respectively. We then manually corrected the errors in the tracking with a guide

from the height-corrected 3D images using a pipeline employing napari [51], and cropped out the region close to the edge so that the remaining region only includes reliably tracked cells. The miss annotations are negligible since the time intervals are significantly shorter than the lifetimes of the cells (average lifetimes were 2.3 days in the hind paw data compared to 0.5 day interval, and 4.0 days in the ear data compared to one day interval). The script outputted the segmented areas of the cells at each time point and their lineages, which was used to build the spatiotemporal graphs. In building the graphs, we determined the cell neighbors from the segmentation images; edges were drawn between cells that share more than one pixel as their periphery. All codes involved in generating the basal layer cell tracks in the ear epidermis were written in Python.

We used two areas of tracking data for the hind paw data, and six areas for the ear data. The total number of cell-frames in the spatiotemporal graphs was 5, 996 for the hind paw data ($\sim$214 unique cells per area per time frame) and 12, 828 for the ear data ($\sim$267 unique cells per area per time frame). We split the graph data into two, the training set and the test set (Fig L in S1 Appendix). The number of cell-frames in the two sets was made to be comparable. The proportion of cells that experience delamination and division in each frame were both 9% in the hind paw epidermis data and both 7% in the ear epidermis data.

For the neighbor fate net imbalance analysis, we applied the same method as previously described [1] to the ear epidermis data as well as to the simulation data.

## Bidirectional spatiotemporal GNN model

The spatiotemporal graphs were created from $N_t$ sequential time frames from the segmented time-lapse images using Deep Graph Library [52]. We first created the cell-contact graphs for each time frame. The neighboring cells $\alpha$ and $\beta$ were connected by two directed edges pointing at each other. We then added directional edges between the same cells in the future and the past as well as their parents and daughter cells in the sequential time frames. The cell feature vector $\boldsymbol{x}_{i,\alpha} \in \mathbb{R}^n$ was assigned to each cell $\alpha$ for the $i$-th time frame. We used the cell area and G1-phase reporter signal as components of the feature vector, which were obtained from the original segmented images. The features were normalized by dividing the values by the maximum values among all the training and test data. When reducing a feature, we set that particular feature to zero in all nodes.

We processed these spatiotemporal graphs with a GNN model using PyTorch. Our first GNN model is a collection of models which consists of a backward temporal edge model $\Phi^{\text{B,edge}}$, a backward temporal node model $\Phi^{\text{B,node}}$, a spatial edge model $\phi^{\text{edge}}$, a spatial node model $\phi^{\text{node}}$, a forward temporal edge model $\Phi^{\text{F,edge}}$, a forward temporal node model $\Phi^{\text{F,node}}$, and a decoder $\psi^{\text{dec}}$.

First, we propagated the information from the future to the past using $\Phi^{\text{B,edge}}$, and then updated the node feature by $\Phi^{\text{B,node}}$. By initializing the node feature $\boldsymbol{a}_{i,\alpha}^{(0)} = \boldsymbol{x}_{i,\alpha}$, the $(k+1)$-th update ($0 \leq k \leq N_t - 2$) of the node feature $\boldsymbol{a}_{i,\alpha}^{(k+1)}$ of cell $\alpha$ in the $i$-th frame ($0 \leq i \leq N_t - 1$) is given by,

$$\boldsymbol{A}_{(i,\alpha)\Leftarrow(i+1,\beta)}^{(k+1)} \quad = \quad \Phi^{\text{B,edge}}\big(\boldsymbol{a}_{i,\alpha}^{(k)}, \boldsymbol{a}_{i+1,\beta}^{(k)}\big), \tag{1}$$

$$\boldsymbol{B}_{i,\alpha}^{(k+1)} \quad = \quad \sum_{(i+1,\beta)\in D(i,\alpha)} \boldsymbol{A}_{(i,\alpha)\Leftarrow(i+1,\beta)}^{(k+1)}, \tag{2}$$

$$\boldsymbol{a}_{i,\alpha}^{(k+1)} \quad = \quad \Phi^{\mathrm{B,node}}\big(\boldsymbol{B}_{i,\alpha}^{(k+1)}, \boldsymbol{a}_{i,\alpha}^{(k)}\big). \tag{3}$$

Here, the subscript $(i, \alpha)$ denotes cell $\alpha$ in the $i$-th frame, and $(i, \alpha)\Leftarrow(j, \beta)$ denotes the edge between cell $\alpha$ in the $i$-th frame and cell $\beta$ in $j$-th frame. We calculated Eq 1 for all the connected pairs of cells in the spatiotemporal graphs. $D(i, \alpha)$ is the set of the daughters of the cell $\alpha$ in the $i$-th frame if the cell $\alpha$ divides in the $i$-th frame; otherwise, $D(i, \alpha)$ is cell $\alpha$ itself in the $(i + 1)$-th frame. We set the features of the cells in the final frame $\boldsymbol{a}_{i,\alpha}^{(k+1)} = \boldsymbol{a}_{i,\alpha}^{(0)}$, which was not updated. Furthermore, for cell $\alpha$ which delaminates or exits the field of view in the $i$-th frame, we set $\boldsymbol{B}_{i,\alpha}^{(k+1)} = 0$.

Second, we calculated the edge features in each time frame using $\phi^{\mathrm{edge}}$, and then updated the cell feature vectors with the edge features using $\phi^{\mathrm{node}}$. By initializing with $\boldsymbol{b}_{i,\alpha}^{(0)} = \boldsymbol{a}_{i,\alpha}^{(N_t-1)}$, the $(l + 1)$-th update of the node feature $\boldsymbol{b}_{i,\alpha}^{(l+1)}$ of cell $\alpha$ in the $i$-th frame is given by,

$$\boldsymbol{H}_{(i,\alpha)\Leftarrow(i,\beta)}^{(l+1)} \quad = \quad \phi^{\mathrm{edge}}\big(\boldsymbol{b}_{i,\alpha}^{(l)}, \boldsymbol{b}_{i,\beta}^{(l)}\big), \tag{4}$$

$$\boldsymbol{I}_{i,\alpha}^{(l+1)} \quad = \quad \mathrm{AGG}\big(\{\boldsymbol{H}_{(i,\alpha)\Leftarrow(i,\beta)}^{(l+1)} : (i, \beta) \in N(i, \alpha)\}\big) \tag{5}$$

$$\boldsymbol{b}_{i,\alpha}^{(l+1)} \quad = \quad \phi^{\mathrm{node}}\big(\boldsymbol{I}_{i,\alpha}^{(l+1)}, \boldsymbol{b}_{i,\alpha}^{(l)}\big). \tag{6}$$

Here, $N(i, \alpha)$ is the set of the neighbor cells of $\alpha$ in the $i$-th frame. In Eq 5, AGG represents either the sum aggregation or the mean aggregation across the set of the neighbor cells. We repeat this process for $N_s$ times to take into account the $N_s$-step neighbor interactions, in which case $0 \leq l \leq N_s - 1$.

Third, we propagated the information from the past to the future using $\Phi^{\mathrm{F,edge}}$, and then updated the node feature by $\Phi^{\mathrm{F,node}}$. By initializing with $\boldsymbol{c}_{i,\alpha}^{(0)} = \boldsymbol{b}_{i,\alpha}^{(N_s)}$, the $(m + 1)$-th update ($0 \leq m \leq N_t - 2$) of the node feature $\boldsymbol{c}_{i,\alpha}^{(m+1)}$ of cell $\alpha$ in the $i$-th frame is given by,

$$\boldsymbol{U}_{(i+1,\alpha)\Leftarrow(i,\beta)}^{(m+1)} \quad = \quad \Phi^{\mathrm{F,edge}}\big(\boldsymbol{c}_{i,\alpha}^{(m)}, \boldsymbol{c}_{i+1,\beta}^{(m)}\big), \tag{7}$$

$$\boldsymbol{V}_{i,\alpha}^{(m+1)} \quad = \quad \boldsymbol{U}_{(i+1,\alpha)\Leftarrow P(i+1,\alpha)}^{(m+1)}, \tag{8}$$

$$\boldsymbol{c}_{i,\alpha}^{(m+1)} \quad = \quad \Phi^{\mathrm{F,node}}\big(\boldsymbol{V}_{i,\alpha}^{(m+1)}, \boldsymbol{c}_{i,\alpha}^{(m)}\big). \tag{9}$$

Here, $P(i + 1, \alpha)$ is the parent of cell $\alpha$ in the $(i + 1)$-th frame if the cell $\alpha$ is born in the $(i + 1)$-th frame; otherwise, $P(i + 1, \alpha)$ is cell $\alpha$ itself in the $i$-th frame. We set the features of the cells in the final frame $\boldsymbol{c}_{i,\alpha}^{(m+1)} = \boldsymbol{c}_{i,\alpha}^{(0)}$, which was not updated. Furthermore, for cell $\alpha$ which pops in the field of view from the outside in the $i$-th frame, we set $\boldsymbol{V}_{i,\alpha}^{(m+1)} = 0$.

Finally, we decoded the cell feature of cell $\alpha$ in the final frame ($(N_t - 1)$th frame) by:

$$\boldsymbol{Y}_{N_t-1,\alpha} = \psi^{\mathrm{dec}}\big(\boldsymbol{c}_{N_t-1,\alpha}^{(N_t-1)}\big). \tag{10}$$

For all the functions $\Phi^{\mathrm{B,edge}} : \mathbb{R}^{2n} \to \mathbb{R}^n, \Phi^{\mathrm{B,node}} : \mathbb{R}^{2n} \to \mathbb{R}^n, \phi^{\mathrm{edge}} : \mathbb{R}^{2n} \to \mathbb{R}^n, \phi^{\mathrm{node}} : \mathbb{R}^{2n} \to \mathbb{R}^n, \Phi^{\mathrm{F,edge}} : \mathbb{R}^{2n} \to \mathbb{R}^n, \Phi^{\mathrm{F,node}} : \mathbb{R}^{2n} \to \mathbb{R}^n, \psi^{\mathrm{dec}} : \mathbb{R}^n \to \mathbb{R}^3$ in our GNN model, we used the multi-layer perceptron (MLP), whose components are (1) $N_{\mathrm{layer}}$ hidden layers

which are respectively composed of a fully-connected layer and a rectified linear unit (ReLU), and (2) an output fully-connected layer.

## Unidirectional spatiotemporal GNN model

For the unidirectional spatiotemporal model, we skipped the backward temporal edge and node models in the bidirectional spatiotemporal model. Hence, we initialized $\boldsymbol{b}_{i,\alpha}^{(0)} = \boldsymbol{x}_{i,\alpha}$ in the spatial edge and node models. In the cell external model, we assigned a null feature vector for the target cells; in the spatial edge model, we initialized $\boldsymbol{b}_{i,\alpha}^{(0)} = \boldsymbol{0}$ and $\boldsymbol{b}_{i,\beta}^{(0)} = \boldsymbol{x}_{i,\beta}$ for the edges from cell $\beta$ to cell $\alpha$ in the $i$-th frame.

To represent the local lineage branch structure, we introduced the next frame behavior (NFB) as a new feature, which encodes the behavior of the next frame of that cell by a one-hot vector, NB ([1, 0, 0]), Del ([0, 1, 0]), or Div ([0, 0, 1]). Since the NFB in the final frame of each network is what we aim to predict, we set the NFB in the final frame to null vector ([0, 0, 0]).

## Training

We trained the GNN models as a three-class classification problem between the three possible fates, NB,Del and Div). Since the proportion of the three cell fates is imbalanced, we used the weighted softmax-cross-entropy loss where the weight of each label was set to the inverse of the proportion of the cell fate in the training data. To minimize the loss, we used Adam optimizer with the learning rate $lr$ = 0.0001. In the training, we input a spatiotemporal graph of $N_t$ sequential time frames obtained from an imaging area to update the parameters of the GNN model. We repeated the update for all the spatiotemporal graphs in a single epoch of the training.

To optimize the number of layers $N_{\text{layer}}$, the number of nodes $N_{\text{node}}$ of a hidden layer of the MLPs, and the dropout rate $p$, we tested the performance of the GNN model by changing these parameters. For this test, we used the simulation data of delamination-induced division setup with NFB and random features. First, we investigated the effect of $N_{\text{layer}}$ by setting $N_{\text{node}}$ = 50 and $p$ = 0. To quantify the performance of the GNN model, we calculated the maximum value of macro-F1 score of the test data during learning for 2000 epochs. We ran 5000 epochs for $N_{\text{layer}}$ = 2, since the learning was exceptionally slow. As shown in Fig Mi in S1 Appendix, $N_{\text{layer}}$ = 1 and $N_{\text{layer}}$ = 2 give comparable performance. Second, we investigated the effect of $N_{\text{node}}$ by setting $N_{\text{layer}}$ = 1 and $p$ = 0. We found that $N_{\text{node}}$ does not significantly affect the performance (Fig Mii in S1 Appendix). Hence, we chose $N_{\text{layer}}$ = 1 and $N_{\text{node}}$ = 50 to minimize the size of the GNN model. Finally, we changed $p$ as shown in Fig Miii in S1 Appendix, and found that $p$ also does not significantly affect the performance, while overfitting is suppressed by increasing $p$ (Fig Miv,v in S1 Appendix). We here chose $p$ = 0.1 since the average performance was slightly better and the standard deviation is smaller compared with the other conditions.

In all the training, we set $N_{\text{layer}}$ = 1, $N_{\text{node}}$ = 50 and $p$ = 0.1, and ran 2000 epochs. We exceptionally ran 5000 epochs for the bidirectional spatiotemporal model with mean aggregation without any feature.

We also tested the effect of data size on the performance (Figs N and O in S1 Appendix). We found that the prediction score and attribution score are sufficiently high when the number of cell-frames used in the training data is above 2000.

The AUC and attribution of each condition were calculated for the GNN model obtained at the epoch at which the model achieves the maximum macro-F1 score [53]. In Fig B in S1 Appendix, we show an example of the learning curves of the cell external model for the hind paw data with NFB and the random feature. The weighted softmax-cross-entropy loss, macro-

F1 score, recall, and precision curves are shown respectively in Fig B in S1 Appendix. The vertical lines in Fig Bi-iv in S1 Appendix indicate the epoch at which the model achieves the best macro-F1 score.

## Attribution method

We used the integrated gradients (IG, [16]) for the attribution. The IG $I_k(\boldsymbol{g})$ of the $k$-th feature for an input subgraph $\boldsymbol{g}$ of a target cell is given by:

$$I_k^f(\boldsymbol{g}) = (X_k - X_k') \cdot \int_0^1 \frac{\partial F^f(\boldsymbol{X}' + \alpha \cdot (\boldsymbol{X} - \boldsymbol{X}'))}{\partial X_k} \, d\alpha. \tag{11}$$

The function $F^f$ is the softmax score of fate $f \in \{$NB, Del, Div$\}$ as a function of the input features calculated by the trained network. $\boldsymbol{X}$ represents the concatenated features of all the cells of $\boldsymbol{g}$, and $X_k$ is the value of the $k$th feature. $\boldsymbol{X}'$ is the baseline, which is the null vector of the same size as $\boldsymbol{X}$. We calculated the IG for three cell fate labels of all the input graphs.

Since the baseline should be neutral for calculating the attribution, the GNN model must be trained for the null graphs to provide equal soft-max scores for three cell fate labels. To this end, we minimized the mean-squared-error (MSE) loss defined as,

$$L_{\text{MSE}} = \frac{1}{N_{\text{sub}}} \sum_f \sum_{n=1}^{N_{\text{sub}}} \left( p_n^{f,\text{null}} - \frac{1}{3} \right)^2 \tag{12}$$

together with the weighted softmax-cross-entropy loss. Here, $p_i^{f,\text{null}}$ is the softmax score of fate $f \in \{$NB, Del, Div$\}$ of the null graph of the $n$-th target cell's subgraph, and $N_{\text{sub}}$ is the number of the subgraphs. Within a single epoch, we first input a spatiotemporal graph to update the parameters using the weighted softmax-cross-entropy loss, and then input the corresponding null graph to update the parameters using the MSE loss. By this learning method, we approximately obtained the neutral baseline softmax scores for null graphs (Fig Pi in S1 Appendix). Also, we confirmed that introducing the MSE loss does not affect the model performance (Fig Pii in S1 Appendix).

In the analysis of the attribution, we pooled each feature of each target cell's subgraph into the relative spatiotemporal position of the feature with respect to the target cell. We calculated the average IG of each category for each cell fate label. With respect to the average IG of NFB, we averaged the IG of each category of NFB only among the cells in the category. Finally, we calculated the mean of the average IG over all the trained GNN models. We also defined the baseline for non-zero signals shown as the orange shades in the attribution plots. The minimum (maximum) value of the range of the baseline is defined as the minimum (maximum) of the model-average of IG subtracted (added) by the standard error among all the pooled random features.

Since $\sum_f F^f(\boldsymbol{X}) = 1$ for any $\boldsymbol{X}$, we have

$$\sum_f I_k^f(\boldsymbol{g}) = 0, \tag{13}$$

To make the plots in Figs 2–4, we pooled the features according to the spatiotemporal position. The pooled features, which we denote as $a$, contain multiple features $k$ in the original

calculation. The attribution score for each pooled feature is evaluated by

$$I_a^f(\boldsymbol{g}) = \frac{\sum_{k \in a} I_k^f(\boldsymbol{g})}{\sum_{k \in a} 1}. \tag{14}$$

Note that the normalization still holds:

$$\sum_f I_a^f(\boldsymbol{g}) = \frac{\sum_{k \in a} \sum_f I_k^f(\boldsymbol{g})}{\sum_{k \in a} 1} = 0. \tag{15}$$

## Numerical simulations

For the numerical data that mimics the dynamics of basal layer dynamics, we took a simplified model of interacting particles that exclude each other through mechanical interactions and undergo stochastic division and elimination events. We placed $N_0$ cells labeled by $\alpha$ in a two-dimensional plane with size $L \times L$ and with periodic boundary conditions and let the cells interact with each other through an interacting potential. The equation of motion reads

$$\dot{\boldsymbol{r}}_\alpha(t) = -\sum_{\beta \neq \alpha} \frac{\partial}{\partial \boldsymbol{r}_\alpha} u(\boldsymbol{r}_\alpha, \boldsymbol{r}_\beta), \tag{16}$$

which is an overdamped kinetics without noise. $\boldsymbol{r}_\alpha$ is the position vector of cell $\alpha$. The repulsive interacting potential has the typical length scale $l$:

$$u(\boldsymbol{r}_\alpha, \boldsymbol{r}_\beta) = \begin{cases} \frac{1}{2} K(|r_\alpha - r_\beta| - l)^2 & |r_\alpha - r_\beta| < l \\ 0 & |r_\alpha - r_\beta| \geq l \end{cases}. \tag{17}$$

The rules of cell division and delamination were implemented by the Monte Carlo method. At the time frame of cell division of cell $\alpha$, a newly born cell $\alpha'$ is generated at a random position within a small distance $d = 0.001 \times L$. Cell delamination is conducted by eliminating a particle instantaneously. In both cases, the position of the cells quickly relaxes to a dispersed state due to the repulsive force between the cells, Eq 16.

The stochastic rules of fate coordination that we tested are:

- Delamination-induced division: delaminating cells are chosen randomly with rate $\lambda$ from the pool of cells that have not yet committed to delaminate or to divide. The chosen cells are committed to delaminate and are assigned a remaining lifetime chosen from a uniform distribution between 32.4 hours and 39.6 hours. At the time point of delamination, one of the six-nearest neighbors of the delaminating cell is randomly chosen, again excluding the cells that have already committed to division or delamination, and is assigned to divide after a randomly chosen remaining lifetime drawn from a uniform distribution between 44.4 hours and 51.6 hours. $N_0 = 612$.

- Division-induced delamination: dividing cells are chosen randomly with the rate $\lambda$ from the pool of cells that have not yet committed to delaminate or to divide. The chosen cells are committed to dividing and are assigned a remaining lifetime randomly chosen from a uniform distribution between 32.4 hours and 39.6 hours. At the time point of division, one of the six-nearest neighbors of the dividing cell is randomly chosen, again excluding the cells that have already committed to division or delamination, and is assigned to delaminate after

a randomly chosen remaining lifetime drawn from a uniform distribution between 44.4 hours and 51.6 hours. $N_0 = 412$.

- Mixed: the two schemes explained above were mixed, with the rate of randomly assigning the delaminating and dividing cells being almost halved so that the overall event rate does not change. $N_0 = 512$.

We used $K = 9$ hours$^{-1}$, $l = 0.125$, and $L = 1$ in all the simulations. For the time steps, we took $\Delta t = 1.2$ hours.

In generating the data for the graph construction, we first prepared $N_0$ points randomly placed inside the box (size $L \times L$) and simulated the time evolution according to Eq 16 by the Euler method for 100 steps to obtain a dispersed cell configuration. Next, we ran the simulation up to 300 steps (360 hours) with both the equation of motion Eq 16 and the stochastic divisions and delaminations, to make sure that the system has reached a steady-state (Fig Q in S1 Appendix). Finally, we ran the simulation for another 300 steps and sub-sampled the time points every 20 steps (24 hours) from this final time series to generate data resembling the ear epidermis. The initial number of cells $N_0$ was changed in the three setups to ensure that the number of cells in the frames at steady-state are roughly the same (around 500, Fig Q in S1 Appendix). The rates of fates $\lambda$ were also tuned for each setup so that the number of events that take place per cell per time frame is comparable with the experiment.

We cropped out the edges and used the data from the points in the center region $0.65\,L \times 0.65\,L$ so that the number of points per frame is roughly the same as the number of cells per frame in the hind paw and ear data (around 210). The neighboring cell network was generated by the two-dimensional Voronoi tessellation, and by whole spatiotemporal network was fed to the GNN learning process in the same way as the experimental data.

## Supporting information

**S1 Appendix.  Fig A: Dependence of the number of iteration of the spatial MP on the performance.** We apply the bidirectional spatiotemporal GNN model with sum or mean aggregation for the hind paw data under various feature conditions. (i) Schematic of the interaction range achieved by the different numbers of iterations of the spatial MP. (ii) The AUC for each cell fate label obtained by averaging over six trained models. Error bar: standard deviation. **Fig B: Learning curves and confusion matrix.** The cell external model with mean aggregation was applied for the hind paw data with the feature condition: (Area, G1 signal, NFB, Random) = (−, −, +, +). The curves of (i) the weighted softmax-cross-entropy loss, (ii) macro-F1 score, (iii) recall and (iv) precision for training and test data, respectively. The vertical dashed lines indicate the epoch at which the model achieves the best macro-F1 score. We evaluated the models with the best macro-F1 score in the main text. As shown in (i), the test loss decreases only slightly due to the low predictability in this setting; higher AUC can be achieved even for high loss within this close-to-random regime. In (v) and (vi), the confusion matrices of the training and test data averaged over 6 models which achieve the best macro-F1 scores are shown. The standard deviation is also shown. Note that the total number of events are fewer than the value reported in Fig L since cells near the spatio-temporal boundaries of the data set were not used. **Fig C: Statistical properties of the hind paw data.** (i,v) Histogram of relative frequency and the boxplot of the normalized area. (ii,vi,vii) Histogram of relative frequency, the boxplot, and the plot of the average of the normalized G1 signal. The error bar in (vii) is the standard deviation. (iii) The correlation matrix between the number of neighboring cells and normalized area. (iv,viii) Histogram of relative frequency, the boxplot of the number of neighboring cells. In (v,vi,viii), the significance obtained by the two-sample two-sided

Kolmogorov–Smirnov test is shown (*: $p < 0.05$, **: $p < 0.01$, ***: $p < 0.001$). In the boxplots, the box shows the quartiles of the dataset while the whiskers show the rest of the distribution. The outliers are defined by the thresholds which are obtained by multiplying the interquartile range by 1.5 and adding (reducing) it to (from) the third (first) quartile. The numbers of cells used for the analysis are 4953 (NB), 517 (Del) and 517 (Div), respectively. **Fig D: AUC of the bidirectional spatiotemporal GNN model and the unidirectional spatiotemporal GNN model with sum aggregation.** (i) The AUC of the bidirectional spatiotemporal GNN model (Bi.) and the unidirectional spatiotemporal GNN (Uni.) with sum aggregation are shown for various feature conditions obtained by averaging the AUC over six trained models. (ii) The AUC of the unidirectional GNN only with random feature and without any feature. Error bar: standard deviation. **Fig E: Effect of NFB on the performance of the unidirectional spatiotemporal GNN model with sum aggregation.** The AUC of the unidirectional spatiotemporal GNN for models under various feature sets obtained by averaging over six trained models. Error bar: standard deviation. **Fig F: Attribution of unidirectional spatiotemporal GNN model with sum aggregation.** The attributions of the unidirectional spatiotemporal GNN model are shown for each feature condition: (Area, G1-phase signal, NFB, Random) = (+, −, +, +) (i) and (−, +, +, +) (ii). The IG averaged over six trained models is shown for each pooled feature. Error bar: standard error. The upper and lower values of the IGs of the random features are shown as the orange zone. **Fig G: The performance of the unidirectional spatiotemporal GNN model without spatial MP.** The AUC of the unidirectional spatiotemporal GNN for models without spatial MP under various feature sets obtained by averaging over six trained models. We ran 5000 epochs for (Area, G1-phase signal, NFB, Random) = (−, −, +, +) because the learning was slower than the other conditions. Error bar: standard deviation. **Fig H: Lifetime distribution.** The lifetime distribution of delaminating and dividing cells as well as the box plots are presented for the (i,ii) hind paw and (iii,iv) ear. In (ii) and (iv), the significance of the two-sample two-sided Kolmogorov–Smirnov test is shown (*: $p < 0.05$, **: $p < 0.01$, ***: $p < 0.001$). In the boxplots, the box shows the quartiles of the dataset while the whiskers show the rest of the distribution. The numbers of cells used for the analysis are 336 (Del in paw), 240 (Div in paw), 252 (Del in ear) and 178 (Div in ear), respectively. **Fig I: Dependence of the number of time frames on the performance.** The cell external model with mean aggregation was applied for the hind paw data with the feature condition: (NFB, Random) = (+, +). (i) The AUC for the different number of time frames. The AUC for each cell fate label obtained by averaging the AUC over six trained models. Error bar: standard deviation. (ii) The attribution of the five-time model. The IG averaged over six trained models is shown for each pooled feature. Error bar: standard error. **Fig J: Neighbor fate net imbalance analysis for the epidermis data and simulations.** Net imbalance of six-nearest neighbor cells around a divided cell and a delaminated cell is plotted for the (i) hind paw and (ii) ear data, as well as for data generated by simulations in the (iii) delamination-induced division setup, (iv) division-induced delamination setup and (v) mixed setup. Error bar: standard error. **Fig K: Segmented images of the hind paw and ear epidermis.** Segmented images are shown for the (i) hind paw and (ii) ear epidermis. Cells undergoing Div, Del, and NB are indicated by red, blue, and yellow circle markers. The cell-contact graph is also shown. **Fig L: Summary of the data sets. Fig M: Dependence of the performance of the GNN models on the hyperparameters.** The cell external model of the unidirectional GNN with mean aggregation is applied to the simulation data from delamination-induced division setup with the feature condition: (NFB, Random) = (+, +). The macro-F1 score averaged over six trained models is plotted against the hyperparameters. Error bar: standard deviation. (i) The effect of $N_{layer}$ is tested with $N_{node} = 50$ and $p = 0$. (ii) The effect of $N_{node}$ is tested with $N_{layer} = 1$ and $p = 0$. (iii) The effect of $p$ is tested with $N_{node} = 50$ and $N_{layer} = 1$. (iv,v) The effect of dropout on suppression

of overfitting is shown. The training curves for $p = 0$ and 0.1 are shown, respectively ($N_{node} = 50$ and $N_{layer} = 1$). **Fig N: Data size dependency on the prediction and attribution: mouse paw data.** The cell external model with mean aggregation is applied to the mouse paw data with the feature condition: (NFB, Random) = (+, +). (i) The AUC for the different number of cell-frames in the training data obtained by averaging the AUC over six trained models. Error bar: standard deviation. (i-v) The attribution is shown for the different numbers of cells: (ii) 731, (iii) 1557, (iv) 2400 and (v) 3015. The result in the main text is for 3015 cells in the training data. The IG averaged over six trained models is shown for each pooled feature. Error bar: standard error. The upper and lower values of the IG of the random feature are shown as the orange zone. **Fig O: Data size dependency on the prediction and attribution: simulation data.** The cell external model with mean aggregation is applied to the simulation data from the delamination-induced division setup with the feature condition: (NFB, Random) = (+, +). (i) The AUC for the different number of cell-frames in the training data obtained by averaging the AUC over six trained models. Error bar: standard deviation. (ii-vi) The attribution is shown for the different numbers of cells: (ii) 479, (iii) 911, (iv) 1913, (v) 2920 and (vi) 3917. The result in the main text is for 2920 cells in the training data. The IG averaged over six trained models is shown for each pooled feature. Error bar: standard error. The upper and lower values of the IG of the random feature are shown as the orange zone. **Fig P: Baseline softmax score.** (i) The baseline softmax score, which is the softmax score for null-graphs, is shown for the four and five-time cell external model with mean aggregation for the hind paw data. The feature condition is (Area, G1 signal, NFB, Random) = ($-$, $-$, +, +). The horizontal dashed line indicates the target baseline softmax score 1/3. (ii) The AUC is shown for the four-time cell external models with and without baseline correction. **Fig Q: Numerical simulations of the homeostatic tissue model.** (i) Time-evolution of the number of cells and (ii) the number of fate events in the simulations of the Del-induced Div model, Div-induced Del model, and the mixed model. We used the data from 15 to 30 days in these simulations for the GNN analyses.
(PDF)

**S1 Video. Workflow of the GNN framework.** 1) acquisition of live-image data (left panel), 2) construction of the spatiotemporal graph by cell segmentation and cell tracking of the live image data (center panel), 3) prediction of the cell fates (right panel). In the right panel, the model used for the prediction is the bidirectional spatiotemporal GNN model with area and G1 signal features, and the cells with filled color are the ones where the predicted fates were correct.
(MOV)

## Acknowledgments

We thank A. Tanaka and H. Tanaka for fruitful discussions. We also thank K. Adachi, Y. Fukai and A. Klein for the critical reading of the manuscript.

## Author Contributions

**Conceptualization:** Takaki Yamamoto, Kyogo Kawaguchi.

**Data curation:** Takaki Yamamoto, Katie Cockburn, Valentina Greco, Kyogo Kawaguchi.

**Formal analysis:** Takaki Yamamoto, Kyogo Kawaguchi.

**Funding acquisition:** Takaki Yamamoto, Katie Cockburn, Valentina Greco, Kyogo Kawaguchi.

**Investigation:** Takaki Yamamoto, Katie Cockburn, Valentina Greco, Kyogo Kawaguchi.

**Methodology:** Takaki Yamamoto, Kyogo Kawaguchi.

**Project administration:** Kyogo Kawaguchi.

**Supervision:** Kyogo Kawaguchi.

**Validation:** Takaki Yamamoto, Kyogo Kawaguchi.

**Visualization:** Takaki Yamamoto, Kyogo Kawaguchi.

**Writing – original draft:** Takaki Yamamoto, Kyogo Kawaguchi.

**Writing – review & editing:** Takaki Yamamoto, Katie Cockburn, Valentina Greco, Kyogo Kawaguchi.

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
