## [Decision Letter · Decision Letter 0]

8 Jul 2022

Dear Dr. Yamamoto,

Thank you very much for submitting your manuscript "Probing the rules of cell coordination in live tissues by interpretable machine learning based on graph neural networks" for consideration at PLOS Computational Biology. As with all papers reviewed by the journal, your manuscript was reviewed by members of the editorial board and by several independent reviewers. The reviewers appreciated the attention to an important topic. Based on the reviews, we are likely to accept this manuscript for publication, providing that you modify the manuscript according to the review recommendations.

While no additional experiments are required, please note that the manuscript will only be accepted provided that the weakness points raised by Reviewer 1 and 3 (e.g. on overfitting and interpretability) are addressed by enhancing or complementing the discussion content.

Sincerely,

Virginie Uhlmann

Associate Editor

PLOS Computational Biology

Jason Haugh

Deputy Editor

PLOS Computational Biology

[LINK]

While no additional experiments are required, please note that the manuscript will only be accepted provided that the weakness points raised by Reviewer 1 and 3 (e.g. on overfitting and interpretability) are addressed by enhancing or complementing the discussion content.

Reviewer's Responses to Questions

**Comments to the Authors:**

Reviewer #1: Summary

The paper presents a framework to discover the cell coordination rules in developing tissues. Specifically, the authors consider a spatiotemporal graph constructed from the segmentation and tracking of the epidermis cells. The graph is used to train a graph neural network (GNN) on the task of cell fate prediction. Using the explanation methods from the field of explainable AI (XAI) the authors show that the trained GNN is able to learn multiple rules which control the multicellular dynamics.

The proposed combination of geometric deep learning and explanation methods is very interesting and as far as I know has not been used for analysing cell coordination in developing tissues before. Enoding the live imaging as a spatiotemporal graph and using the techniques proposed by the authors constitute a generic framework for analysis of cell interactions.

Strengths

1. The work proposes a general purpose framework for analysing interactions in spatiotemporal graphs of developing tissues.

2. It shows that even without the node features, the structure of the spatiotemporal graph can be used to accurately predict the cell fate with GNN.

3. The authors encode hypothetical cell coordination rules as additional node features and use the explanation methods to show which rules/features are most relevant for cell fate prediction.

4. The proposed method can detect previously known as well as new cell interaction rules.

5. Extensive ablation study of the method is shown in the paper and the appendix.

6. Additional results on the simulated data confirm the effectiveness of the presented method for extracting the cell interaction laws.

7. The paper is well-written and easy to follow. The source code and datasets are provided to reproduce the experiments.

Comments/Questions

1. The claim that the GNN is significantly better than a random guess at the cell fate classification is not strongly supported. Fig. S2 D shows the precision score is slightly above 0.8 for the NB class and between 0.1-0.2 for the Div/Del classes (test set). Given the 1:1:8 ratio between the classes (Div/Del/NB) the GNN performance is only slightly better than random.

The original claim would be more convincing if the authors show quantitative comparison with a simple baseline (e.g. shallow model trained from the node features and the aggregated features of immediate neighbours) as well as other types of GNN architectures (e.g. GCN [1], GAT [2]).

2. The learning curves in Fig. S2 A show that the model is overfitting very quickly and the checkpoint chosen based on the best F1 score is in the overfitting territory. Have the authors considered more aggressive regularization (e.g. higher dropout rate, weight decay)?

3. It would be interesting to see if the GNNs trained on the paw epidermis generalises well (without re-training) on the ear epidermis and vice versa. This could show if the rules learned by the GNN are transferable to a slightly different tissue. One could also try an external dataset such as epithelial cells of the Drosophila wing disc [3].

4. In order to use the integrated gradients (IG) for the attribution the authors trained the GNN with an additional MSE loss. This might negatively impact the original task of the cell fate prediction. What was the weighting factor (\\alpha) for the MSE term, assuming the final loss is of the form L_{CE} + \\alpha L_{MSE}? Have the authors considered simpler attribution methods which would not require training GNNs for null graphs (e.g. SmoothGrad [4], Gradient x Input [5])?

[1] T. Kipf et al. Semi-supervised classification with graph convolutional networks.

[2] P. Velickovic et al. Graph attention networks.

[3] J. Funke et al. A Benchmark for Epithelial Cell Tracking.

[4] D. Smilkov et al. SmoothGrad: Removing noise by adding noise.

[5] A. Shrikumar et al. Learning Important Features Through Propagating Activation Differences.

Reviewer #2: The authors present an approach for identifying associations between a cell's fate and features of (other) cells in its temporal and spatial vicinity. The approach is based on a spatiotemporal graph neural network (GNN) trained to predict cell fate in the final time frame, together with an attribution method (integrated gradients) that allows to identify input features relevant for the GNN's predictions.

The authors apply this framework to mouse skin cell lineage trees, and find that besides a previously reported association of cell delamination with preceding division of a spatially neighboring cell, delamination of a cell also suppresses delamination in the vicinity in subsequent timeframes, and, in densely packed tissue, cell division is associated with preceding delamination of a neighboring cell.

The proposed engineering of a framework from existing methodology (GNN architecture, IG) is well motivated, as is the training setup and the investigated model ablations. The results are convincing. In particular, while the accuracy of the GNN in terms of predicting cell fate is far from perfect / only slightly better than random in some cases (see Fig S2 F), it is still possible for IG to detect associations that are intuitively plausible, as opposed to previously reported methodology (neighboring imbalance analysis) that is less sensitive and hence only detects some but not all of the above associations.

The authors claim that their GNN architecture is novel in that it captures both spatial and temporal links. However, such kind of architecture has been used before, albeit for the cell tracking problem and not for cell fate prediction on an existing lineage tree (see e.g. https://arxiv.org/pdf/2202.04731.pdf). Nevertheless, this related line of work should be mentioned to contextualize the claimed novelty of the proposed GNN architecture.

A minor request: At some points in the manuscript references are cited not at the first mention of the respective method or finding, but somewhere later in the text (e.g., "the attribution method", and later specifically "IG"), which can be very easily fixed by just moving up citations respectively.

Reviewer #3: The Authors propose using Graph Neural Networks (GNN) as an effective and interpretable tool to analyze cell-to-cell interactions.

In particular:

-The Authors built a spatio-temporal cell adjacency graph from live-cell time-lapses and created a cell fate classification task.

-The Authors performed extensive experiments to show the strength of their GNN framework, an in-depth features analysis, and ablation studies.

-The Authors used integrated gradients (IG) as a mathematical tool to analyze features importance and interpret the GNN results.

Strengths:

Overall the manuscript showcases an interesting way of approaching biological questions, such as discovering multicellular dynamics rules. Using machine learning not only as an “automation” tool but as an instrument to understand complex interaction is a fascinating and promising (although challenging) direction.

In particular:

- the careful and detailed use of IG as a tool for dissecting the relevance of the features and time points.

- a large number of carefully crafted experiments and ablation studies.

- the manuscript reads easily, and the methods are detailed and explain all relevant implementation details well.

Weaknesses:

-The experiments reported in the manuscript seem to show that the spatial MP has very little influence on the model AUC performance, i.e., it does not contribute to the GNN predictions.

This is suggested in Fig. S7 Appendix S1, where the results without spatial MP seem to be almost as good as in Fig. 2. But also in Fig. S1 Appendix S1, where it is shown that changing the Spatial interaction range does not affect the classification AUC.

Would a “standard” (for example, a multilayer perceptron) classifier with cell (no neighbor features) and temporal features (like NFB at Nt=-1 (cell parent), -2 (cell grandparent), -3..) have the same classification score? If so, what is the additional benefit of directly encoding the spatial-MP and forward-MP in a GNN compared to merely using the NFB features in a standard classifier?

- The seemingly little impact of the neighbor features mentioned above is not only observed in the AUC. The IG also shows minimal impact on neighbor features when the model is exposed to target features (Fig. 2), while the neighbor features relevance is high when target features are artificially “zeroed” Fig. 3 and 4. Thus, if my interpretation of the results is correct, the discovery of processes such as “Delamination-induced division” and “Lateral inhibition of delamination” (that depends strongly on the spatial neighborhood context) is not possible from the full unidirected model results but requires the model to be ad-hoc modified for this discovery.

But, in future applications, adapting the model to reveal more complex interaction processes is not straightforward and might limit the applications of the proposed framework.

-The Authors emphasize that their method is inherently more “interpretable” than alternative approaches (for example, CNN-based approaches), mainly because GNNs directly incorporate the cell spatio-temporal relationship.

But, in the manuscript, the primary source of interpretability comes from the features impact analysis rather than the graph structure. As it stands in the manuscript, there is no clear advantage of using message passing. Have the Authors thought about using attention-based GNN? An analysis of the attention pattern could lead to a more atomized understanding of which neighbor cells are important for the model to make accurate predictions.

-The Authors, between line 191 and line 197, state that the NFB features are the key ingredients (in the absence of backward-time MP) for the model performance. But in Fig. S5 Appendix S1, the second column from the left (Area, G1, NFB, rand: ++-+) suggests that even without NFB, the unidirectional model can achieve good performance using only Area and G1. In the same spirit of my first question, how powerful would a classifier be with Area and G1 as the only two features (without any MP)?

Could the drastic reduction in performance you show in Fig. 2C (Area, G1, NFB, rand: ---+) compared to Fig. 1E (--) be a consequence of the random feature rather than the absence of the NFB?

- The Authors should discuss (if possible) how and how much the error rate impacts the confidence in their biological findings. For example, is an AUC of 0.6 (such as for the Delamination detections) enough to understand the system understudy fully? In this sample case, a systematic error in the network predictions might hide significant interactions and bias the rule extracted.

What could be improved to reduce the risk of such systematic error and improve performance? For example, would a more extensive and complex set of cell features help? Or would it make results harder to interpret due to possible features cross-correlation?

Recommendation:

Although I have expressed some doubts and have several questions about the proposed framework and experimental results (details in the “Weaknesses” section of my review), my overall feedback on the manuscript is positive, and I encourage the Authors to provide a revised version.

**Have the authors made all data and (if applicable) computational code underlying the findings in their manuscript fully available?**

Reviewer #1: Yes

Reviewer #2: Yes

Reviewer #3: Yes

PLOS authors have the option to publish the peer review history of their article (what does this mean?). If published, this will include your full peer review and any attached files.

Reviewer #1: **Yes: **Adrian Wolny

Reviewer #2: No

Reviewer #3: **Yes: **Lorenzo Cerrone

Figure Files:

Data Requirements:

Reproducibility:

References:

---

## [Editor Report · Decision Letter 1]

9 Aug 2022

Dear Dr. Yamamoto,

We are pleased to inform you that your manuscript 'Probing the rules of cell coordination in live tissues by interpretable machine learning based on graph neural networks' has been provisionally accepted for publication in PLOS Computational Biology.

Best regards,

Virginie Uhlmann

Associate Editor

PLOS Computational Biology

Jason Haugh

Deputy Editor

PLOS Computational Biology

Thank you for thoroughly addressing the reviewers' points.

---

## [Editor Report · Acceptance letter]

29 Aug 2022

PCOMPBIOL-D-22-00629R1 

Probing the rules of cell coordination in live tissues by interpretable machine learning based on graph neural networks

Dear Dr Yamamoto,

I am pleased to inform you that your manuscript has been formally accepted for publication in PLOS Computational Biology. Your manuscript is now with our production department and you will be notified of the publication date in due course.

With kind regards,

Zsofia Freund
